# Sodium Ion Pre-Intercalation of δ-MnO_2_ Nanosheets for High Energy Density Aqueous Zinc-Ion Batteries

**DOI:** 10.3390/nano13061075

**Published:** 2023-03-16

**Authors:** Yuanhao Ding, Weiwei Xue, Kaihao Chen, Chenghua Yang, Qi Feng, Dezhou Zheng, Wei Xu, Fuxin Wang, Xihong Lu

**Affiliations:** 1School of Applied Physics and Materials, Wuyi University, Jiangmen 529020, China; 2MOE of the Key Laboratory of Bioinorganic and Synthetic Chemistry, The Key Lab of Low-Carbon Chem & Energy Conservation of Guangdong Province, School of Chemistry, Sun Yat-sen University, Guangzhou 510275, China

**Keywords:** δ-MnO_2_, pre-intercalation, aqueous zinc-ion batteries, cathode

## Abstract

With the merits of low cost, environmental friendliness and rich resources, manganese dioxide is considered to be a promising cathode material for aqueous zinc-ion batteries (AZIBs). However, its low ion diffusion and structural instability greatly limit its practical application. Hence, we developed an ion pre-intercalation strategy based on a simple water bath method to grow in situ δ-MnO_2_ nanosheets on flexible carbon cloth substrate (MnO_2_), while pre-intercalated Na^+^ in the interlayer of δ-MnO_2_ nanosheets (Na-MnO_2_), which effectively enlarges the layer spacing and enhances the conductivity of Na-MnO_2_. The prepared Na-MnO_2_//Zn battery obtained a fairly high capacity of 251 mAh g^−1^ at a current density of 2 A g^−1^, a satisfactory cycle life (62.5% of its initial capacity after 500 cycles) and favorable rate capability (96 mAh g^−1^ at 8 A g^−1^). Furthermore, this study revealed that the pre-intercalation engineering of alkaline cations is an effective method to boost the properties of δ-MnO_2_ zinc storage and provides new insights into the construction of high energy density flexible electrodes.

## 1. Introduction

Given the rapid depletion of traditional fossil fuels and the high rate of demand for renewable energy, it is imperative to explore large-scale energy storage devices [1,2]. As the most commercially mature electrochemical energy conservation equipment, lithium-ion batteries are already extensively and deeply studied [3,4,5]. However, the potential insecurity, scarce lithium resources and high costs hinder their further development. In the past decade, many aqueous rechargeable batteries have emerged as a result of their much higher conductivity than organic electrolytes. Among them, aqueous zinc ion batteries (AZIBs) have broad application prospects in energy storage by virtue of their abundant zinc resources, inexpensive, high theoretical specific capacity and safety [6,7,8]. The AZIBs are assembled by cathode, anode and diaphragm, in which the cathode material is a vital part of the AZIBs, which largely determines the performance of the whole battery, because most of the currently reported cathode materials for AZIBs possess relatively weak capacity (200–300 mAh g^−1^), while the theoretical capacity of zinc anode is up to 820 mAh g^−1^ [9,10,11]. Hence, the development of cathode materials with high zinc storage capacity and long cycle life is important for the commercialization of AZIBs.

Among various AZIBs cathode materials, manganese-based materials have been considered as high-potential cathodes for AZIBs as a result of their low cost, abundant storage, multivalent state (Mn^0^, Mn^2+^, Mn^3+^, Mn^4+^, Mn^7+^), high theoretical capacity (308 mAh g^−1^), high voltage platform (1.4 V vs. Zn^2+^/Zn), and environmental friendliness [12,13,14,15]. MnO_6_ octahedra are the base structural unit of MnO_2_, and the different ways of their connection lead to a number of crystal structures. For instance, α-MnO_2_, whose basic structural unit consists of coplanar MnO_6_ octahedral double chains, shares the top angle with its adjacent double chains to form 2 × 2 and 1 × 1 tunneling structures that allow the hosting of various cations or ligand ions [16,17]. The δ-MnO_2_ layers are only linked by slight Van der Waals interaction, resulting in a large intra-layer space with many ion-embedded active sites [18,19,20]. Among them, the tunneling or lamellar structures of MnO_2_ units are able to reversibly intercalate/deintercalate Zn^2+^ and have attracted extensive research. Xu et al. used the microemulsion method to compose α-MnO_2_ as the cathode of an aqueous phase zinc ion battery and obtained a capacity of 210 mAh g^−1^ at 1 A g^−1^ with few decreases of capacity over 50 cycles [21]. Chen et al. prepared β-MnO_2_ via a solvothermal approach and found that β-MnO_2_ was able to reversibly insert/extract Zn^2+^ by phase transition during primary discharge to form Zn-buserite phase, which was used as the cathode of AZIBs and obtained a capacity of 225 mAh g^−1^ at 0.2 A g^−1^ and maintained 94% capacity over 2000 cycles [22]. Although tunneled β-MnO_2_ possesses excellent thermodynamic stability, favorable Zn^2+^ diffusion ability and architectural sustainability, the tunneling is too narrow and the diffusion capacity of Zn^2+^ is insufficient, limiting the zinc storage capacity of MnO_2_. The layered δ-MnO_2_ with larger interlayer spacing facilitates the Zn^2+^ storage and transport [20] and is an ideal MnO_2_ cathode material. However, the structure can collapse with repeated insertion/extraction of Zn^2+^, leading to a decay in the cycling stability of the electrode material.

To inhibit the deterioration of zinc storage capacity due to layer structure breakdown, pre-insertion of cations or molecules has been shown to be a very practical approach. Pre-inserted cations or molecules can broaden the distance between the layers of MnO_2_ and serve to reduce the volume change during zinc ion insertion/extraction. Liu et al. embedded Cu^2+^ ions between δ-MnO_2_ layers by the solvothermal process, which was used to prepare high-performance CMO cathodes for AZIBs, taking a total of 12 h. The fabricated battery based on CMO cathodes acquired a specific capacity of 175 mAh g^−1^ at 2.0 A g^−1^ [23]. Yang et al. grew ultrathin δ-MnO_2_ nanosheets on a self-constructed flexible carbon film by a solvent method, with interlayer spacing expanded by K^+^ intercalation. The fabricated KMO cathode obtain a capacity of 190 mAh g^−1^ at 3 A g^−1^ [24]. Nevertheless, the capacity of δ-MnO_2_ still has potential for enhancement, as well as the fabrication procedure is relatively complex and takes extra time. Exploring simple methods to obtain higher zinc storage performance of MnO_2_ materials is not only important but also challenging.

Herein, we developed a simple Na^+^ pre-intercalation δ-MnO_2_ method for in situ deposition of Na-MnO_2_ nanosheets on flexible substrate carbon cloth (Na-MnO_2_) through a simple water bath method for its debonding. The pre-intercalated Na^+^ can broaden the layer-to-layer distance and serve as a column support, which greatly improves the ion transport rate and stability of MnO_2_ nanosheets. In consequence, the fabricated Na-MnO_2_//Zn battery with Na-MnO_2_ cathode and Zn plate anode obtained an upper specific capacity of 251 mAh g^−1^ at 2 A g^−1^, as well as very fine rate performance (96.9 mA h g^−1^ at 10 A g^−1^). In addition, the Na-MnO_2_//Zn battery still has 62.5% of the initial capacity after 500 cycles, as well as almost 100% Coulomb efficiency. The prepared battery obtained a remarkable energy density (289 Wh kg^−1^) and peak power density (0.967 kW kg^−1^), which were calculated based on the mass of the active material. This work offers an innovative approach to the development of high-performance flexible manganese-based AZIBs.

## 2. Experimental Section

### 2.1. Synthesis of MnO_2_ Electrodes

All reagents used in this experiment were analytically pure without purification. KMnO_4_ with a measured concentration of 0.15 M was dispersed in 50 mL of deionized water, agitated for a period of time and then drop into 0.3 mL pure H_2_SO_4_ (18.4 M). After fully dissolved, the above mixture was kept in a water bath at 85 °C, and then cleaned carbon cloth (3 × 3 cm), which was ultrasonically treated with deionized water, acetone and ethanol for 5 min and then vacuum dried, was put into the above solution and reacted for 1 h. The reacted carbon cloth was taken out and dried in a vacuum furnace. Finally, the synthesized sample was put into the muffle oven and heated up to 350 ℃ at 5 ℃/min for 1 h to yield MnO_2_ electrodes.

### 2.2. Synthesis of Na-MnO_2_ Electrodes

Based on the above synthesis, 18.75 mM of NaCl was added to the precursor solutions of MnO_2_ to obtain Na^+^ pre-intercalation MnO_2_ electrode materials, namely Na-MnO_2_.

### 2.3. Fabrication of AZIBs

A commercial zinc plate (12 mm dia., Xingxing Shang Metal Material Co., Dongguan, China) was assembled as the anode, a synthetic MnO_2_ and Na-MnO_2_ materials (12 mm diameter) as the cathode, 1 M ZnSO_4_ as the electrolyte, and a glass fiber filter as the diaphragm to form a stainless steel CR2016 coin type AZIBs.

### 2.4. Materials Characterization

X-ray diffraction analyzers (XRD, Bruker Germany, Karlsruhe, Germany) were used to analyze the crystalline architecture of prepared samples. Field emission scanning electron microscopy (SEM, Sigma500, ZEISS, Oberkochen, Germany) was applied to study the morphology of the prepared samples, and energy dispersive X-ray spectroscopy (EDS, INCA 300, Oxford Instruments, Abington, UK) was utilized to observe the elemental distribution. The microstructure of the material was investigated by transmission electron microscopy (TEM, JEM-F200, JEOL, Tokyo, Japan). Determination of the chemical composition of material surfaces using Raman spectroscopy (LabRAM HR Evolution, HORIBA, Kyoto, Japan) and X-ray Photoelectron Spectroscopy (XPS, ESCALAB Xi+, Thermo VG, London, UK).

### 2.5. Electrochemical Performance Characterization

The electrochemical performance test was performed using an electrochemical workstation with the model CHI660e manufactured by Shanghai C&H, Shanghai, China. The specific capacity, rate performance, and energy density of the fabricated batteries were characterized by choosing the test methods of galvanostatic charge/discharge, cyclic voltammetry and electrochemical impedance spectroscopy (GCD, CV, EIS) (CHI660E, CH Instruments, Inc., Austin, TX, USA).

## 3. Results and Discussions

The pure MnO_2_ nanosheets were synthesized through a simple water bath method for in situ grown on flexible substrate carbon cloth. As seen in Figure 1a,b, pure MnO_2_ nanosheets were homogeneously coated on the fibers of flexible substrate carbon cloth. The pure MnO_2_ consisted of thin nanosheets with flower-like morphology, an average particle size of ~90 nm, and a thickness of 10 nm. Figure 1c,d shows SEM images of MnO_2_ after Na^+^ pre-intercalation. From Figure 1a–d, the Na^+^ pre-intercalation has little effect on the morphology of MnO_2_, which still maintains the uniform nanosheet morphology and good compactness. To further determine the location where the sodium ions are present, they were investigated by TEM. Appendix A shows that the MnO_2_ sample possesses a nanosheet morphology, one that is in agreement with the SEM findings. HRTEM of the MnO_2_ sample is shown in Appendix A, which clearly displays the lattice spacing of 0.53 nm, corresponding to (001) crystal plane of δ-MnO_2_ [25]. The Na-MnO_2_ sample also possesses a nanosheet morphology displayed in Figure 1e, indicating that the Na^+^ pre-intercalation does not disrupt the morphological architecture of the material. The lattice spacing of the material was observed to become larger using high-resolution transmission electron microscopy (HRTEM) with a value of 0.58 nm (Figure 1f), demonstrating that Na^+^ ions were successfully pre-intercalated into the interlayer of δ-MnO_2_. Compared with pure MnO_2_, Na^+^ pre-intercalation can increase its planar spacing, which may facilitate the migration of Zn^2+^ in Na-MnO_2_. Immediately after that, the energy spectrum analysis department was performed, and Figure 1h–j shows the elemental distribution maps obtained from the selected closed regions in Figure 1g, which were able to effectively detect sodium elements, where each element (O, Mn, Na) is uniformly distributed on the nanosheets, further indicating that the sodium ions are uniformly embedded in the interlayer of MnO_2_.

To further study the influence of Na^+^ pre-intercalation on the crystal structure, X-ray diffraction (XRD) measurements were conducted on the pure MnO_2_ and Na-MnO_2_. From Figure 2a, it can be seen that the characteristic peaks of the pure MnO_2_ coincided with (001) and (002) planes of δ-MnO_2_ (JCPDS: 80-1098), while no other peaks were detected, implying a relatively high purity of the samples [26]. Since the δ-MnO_2_ is directly grown in situ on the flexible substrate carbon cloth, there is a graphitic carbon peak at about ~25° on both MnO_2_ and Na-MnO_2_ samples, which overlaps with the (002) characteristic peak of δ-MnO_2_, so there is a broad peak appears at this position. Moreover, the pre-intercalation of Na^+^ shows almost no change in the peak pattern, but the diffraction peak of Na-MnO_2_ in the (001) plane is obviously shifted to the left. According to the Bragg equation (2dsin θ = nλ) [27], it is known that the crystal plane spacing increases, which again proves the successful pre-intercalation of Na^+^. The structural features of δ-MnO_2_ were further investigated by Raman spectroscopy (Figure 2b). V_1_ (620–635 cm^−1^), V_2_ (560–575 cm^−1^) and V_3_ (490–510 cm^−1^) represent the three characteristic peaks of δ-MnO_2_. Among them, V_1_ bond is the symmetric stretching vibration of the MnO_6_ group (Mn-O); V_2_ bond corresponds to (Mn-O) stretching within the base plane of the MnO_6_ sheet; V_3_ bond corresponds to (Mn-O) stretching vibration [28,29]. From the Figure 2b, it can be seen that the planar stretching vibration peak moved from 567 cm^−1^ to 564 cm^−1^, and the intensity of the peak decreased, this change is thought to be due to the fact that the trigonal sites between the layers of the MnO_6_ substrate are the most accessible locations prior to the insertion of the Na^+^ and the pre-intercalation of Na^+^ between the MnO_6_ substrates resulting in a change in the material framework [25,28]. The chemical composition of the material and the valence states of the elements were measured by XPS. Figure 2c displays the survey spectrum of Na-MnO_2_ and pure MnO_2_ samples. Due to the relatively small sodium content of the pre-intercalation, the characteristic peak of sodium elements is not obvious. However, the molar content of the Na element in the Na-MnO_2_ sample is about 0.18% from the result of the XPS measurements, while the MnO_2_ electrode failed to detect the Na element, thus also proving the successful pre-intercalation of sodium ions. According to the relevant literature, the average oxidation state (AOS) of Mn is calculated as follows: AOS = 8.956 − 1.126Δ*Es*, where Δ*Es* is the binding energy difference between the two-state Mn 3s peaks [30]. The distances (Δ*Es*) of the double-fitted Mn 3s peaks for the MnO_2_ and Na-MnO_2_ samples were measured to be 4.85 and 4.9 eV (Appendix A), corresponding to the average valence state of 3.49 and 3.44 for Mn, respectively. The pre-intercalation of sodium ions in δ-MnO_2_ is accompanied by the entry of some electrons into the conduction band, giving rise to a partial reduction of Mn^4+^ to Mn^3+^. This particular electron leaving domain leads to the widening of the layer-to-layer spacing of Na-MnO_2_ [31], which is according to the XRD results described above. In high-resolution Mn 2p XPS spectra we can see two typical peaks at about 642.4 eV and 654.2 eV, coinciding with the Mn 2p_3/2_ and Mn 2p_1/2_ spin-orbit splitting seams (Figure 2d). In comparison with MnO_2_, Mn^3+^ can be clearly observed at a characteristic binding energy of about 641.5 eV, which is attributed to the pre-embedding of sodium ions, leading to an increase in the lattice spacing and favoring the ion diffusion during electrochemistry [31,32]. From Figure 2e, it can be observed that the O 1 s spectra exhibit two distinct peaks, corresponding to the Mn–O–Mn bond in tetravalent oxide (at about 530.0 eV) and Mn–O–H bond in hydrated trivalent oxide (at about 531.3 eV), respectively. By comparing the curve integral area of these two samples, it is evident that the percentage of Mn–O–Mn increased from 3.6 to 5.8 after the pre-intercalated Na^+^. This phenomenon can be attributed to the pre-intercalated Na^+^ facilitating the formation of Mn–O–Mn bonds [33,34]. Figure 2f shows the core-level Na 1s spectra of the Na-MnO_2_ sample (the red curve represents the fitted curve). Even though the Na elemental is present at a low level in the sample, the presence of the characteristic peak of Na 1s at 1070.83 eV can be obviously seen in the figure, which again proves the presence of elemental Na.

To further explore the impact of the Na^+^ pre-intercalation layer on the zinc storage properties of these materials, the pure MnO_2_ and Na-MnO_2_ were assembled as cathodes into coin-shaped AZIBs with 1M zinc sulfate as the electrolyte and zinc plate as the anode. As shown in Figure 3a, MnO_2_ and Na-MnO_2_ electrode materials were measured at the scanning rate of 10 mV s^−1^ with a high potential window of 0.8–1.9 V. The cyclic voltammetry (CV) curves of both batteries display similar patterns, but the CV curves integral area of the Na-MnO_2_//Zn battery is larger than that of MnO_2_//Zn, and the redox peak is slightly higher than that of MnO_2_//Zn. This implies that there is sufficient oxidation and reduction in AZIBs for their prepared Na-MnO_2_ cathode materials [12]. Furthermore, there was a clear voltage plateau at 1.2 V for the GCD curves of MnO_2_//Zn and Na-MnO_2_//Zn batteries, corresponding to the reduction peaks (1.18 V) from the CV curves (Figure 3b) [35]. Compared with the MnO_2_//Zn battery, the voltage gap of the Na-MnO_2_//Zn battery is narrower, further indicating that the pre-intercalation strategy of sodium can effectively improve redox reaction activity of δ-MnO_2_ cathode materials [36,37,38]. As can be seen in its rate capacity plot (Figure 3c), the Na-MnO_2_//Zn battery obtained specific capacities of 236.3, 134.1, 118.8, 106.9 and 96.9 mAh g^−1^ at current densities of 2, 4, 6, 8, 10 A g^−1^, respectively, significantly higher than those of the MnO_2_//Zn battery, owing to the pre-intercalation of sodium greatly improving the transport rate of zinc ions. Figure 3d is a graph of its cycle life and it can be seen that the Na-MnO_2_//Zn battery delivers a discharge-specific capacity of 152.8 mAh g^−1^ initial capacity in the first cycle at a current density of 8 A g^−1^. Over the next 50 cycles, the specific capacity gradually reduced, probably attributed to a slight dissolution of the cathode material on the surface and irreversible reactions occurring within the first few cycles [10,39,40]. After 50 cycles, the capacity of the Na-MnO_2_//Zn battery gradually stabilized at about 106 mAh g^−1^ at 8 A g^−1^. Moreover, after 500 cycles, the Na-MnO_2_//Zn battery still achieved a discharge-specific capacity of 96 mAh g^−1^ with nearly 100% Coulomb efficiency, while that of the MnO_2_//Zn battery was only 35 mAh g^−1^. The comparative results show that the pre-intercalation of Na^+^ resulted in a significant improvement in the cycling stability of the Na-MnO_2_//Zn battery. More importantly, the fabricated Na-MnO_2_//Zn battery exhibited a significant energy density of 289 Wh kg^−1^ while the power density was 967 W kg^−1^. As seen in Figure 3d, the energy storage performance of the fabricated Na-MnO_2_//Zn is better than other newly reported energy storage devices (Table 1).

To further explore the material structure-performance relationship, the CV tests were conducted for the prepared Na-MnO_2_//Zn battery and MnO_2_//Zn battery at various scan rates, illustrated in Figure 4a and Appendix A. The measured peak current (*i*) and scan rate (*v*) obey the power-law relationship equation: *i* = *av^b^*, where the values *a* and *b* are empirical constants [45,46]. In general, as the value of b tends to 1, it means that the capacitance dominates the cycling procedure, and as b nears 0.5, it means that the discharge/charge procedure is controlled by diffusion. The *b* values of the Na-MnO_2_ were calculated to be 0.72 and 0.59 corresponding to the positions of peaks 1 and 2 (Figure 4a). Compared to other oxides, this material has a high electric capacity. For the MnO_2_ electrode, the corresponding *b* values are 0.71 and 0.48 (Appendix A). This indicates a combined regulation of the electrochemical reactions on these peaks by the ion diffusion and capacitive contribution. The higher *b* values for the Na-MnO_2_ electrode indicated that the capacitive characteristics of the Na-MnO_2_ electrode are superior to that of the MnO_2_ electrode. To discriminate the percentage of capacitance in the total capacity, the contribution percentage was quantified by the following equation: *i* = k_1_*v* + k_2_*v*^1/2^, while k_1,_ k_2_ are the defining constants, *i* is the current response at electrostatic potential (V), and *v* is the scan rate. The k_1_*v* represents the capacitance control and k_2_*v*^1/2^ refers to the contribution of diffusion control [47]. The capacitance contributions of 65% and 91% were calculated for the MnO_2_ and Na-MnO_2_ electrodes, respectively, with a sweep rate of 10 mV s^−1^, as well as the shaded areas in Appendix A and Figure 4c for demonstration. From Appendix A and Figure 4c, the capacitance contributions are 49%, 56%, 67%, 74%, 82%and 91% at 1, 2, 4, 6, 8 and 10 mV s^−1^, which is higher than the equivalent conditions for the MnO_2_ electrode. This indicates that the sodium ion pre-intercalation layer δ-MnO_2_ not only promotes reversible Zn^2+^ deem bedding but also contributes to a higher large capacitance capacity. The above results confirm that the proportion of capacitance control in the overall capacity increases as the scan rate increase and that the reaction kinetics of Na-MnO_2_//Zn battery is faster because of the enhanced Zn^2+^ deem bedding kinetics at the Na-MnO_2_ electrode, which is consistent with the excellent rate performance exhibited in this work. Meanwhile, the excellent cycling life of the Na-MnO_2_//Zn battery is achieved due to the synergistic contribution of Zn ion diffusion and electrode capacitive behavior. To deeply understand the reaction kinetics process of Na-MnO_2_ and MnO_2_ electrodes, an electrochemical impedance spectroscopy (EIS) measurement was done, illustrated in Figure 4d. The semicircle in the high-frequency region has an effect on the charge transfer resistance at the electrode-electrolyte interface, and the straight line in the low-frequency region is related to ion diffusion. The charge transfer resistance of the Na-MnO_2_ electrode (139 Ω) is much lower than that of the MnO_2_ electrode (254 Ω) [28]. Therefore, pre-intercalation Na^+^ in δ-MnO_2_ can effectively reduce the charge transfer resistance and accelerate the diffusion kinetics of Zn^2+^. In addition, the slope of Na-MnO_2_ is bigger than that of MnO_2_ from the data in the low-frequency region mainly due to the finite length diffusion, indicating that pre-intercalation of sodium can widen the interlayer spacing and increase the electronic and ionic conductivity.

## 4. Conclusions

In summary, we developed an interlayer modulation strategy based on the water bath method to prepare high-performance Na-MnO_2_ electrode materials. Thanks to the pre-intercalation of sodium ions into the interlayer, MnO_2_ layer spacing is expanded, which is more favorable for the transport of Zn ions. The assembled Na-MnO_2_//Zn battery achieved a maximum capacity of 251 mAh g^−1^ at 2 A g^−1^. Additionally, the battery showed a superior rate capability of 96 mAh g^−1^ at 8 A g^−1^ and an acceptable cycling performance with a capacity retention of more than 62.5% over 500 cycles. Moreover, the Na-MnO_2_//Zn battery exhibited a high energy density of 289 Wh kg^−1^ and a peak power density of 0.967 kW kg^−1^. The ionic pre-intercalated strategy can effectively improve the capacity and stability of electrode materials, and this strategy can be extended to the design and synthesis of layer materials for other flexible rechargeable aqueous batteries.

## Figures and Tables

**Figure 1 nanomaterials-13-01075-f001:**
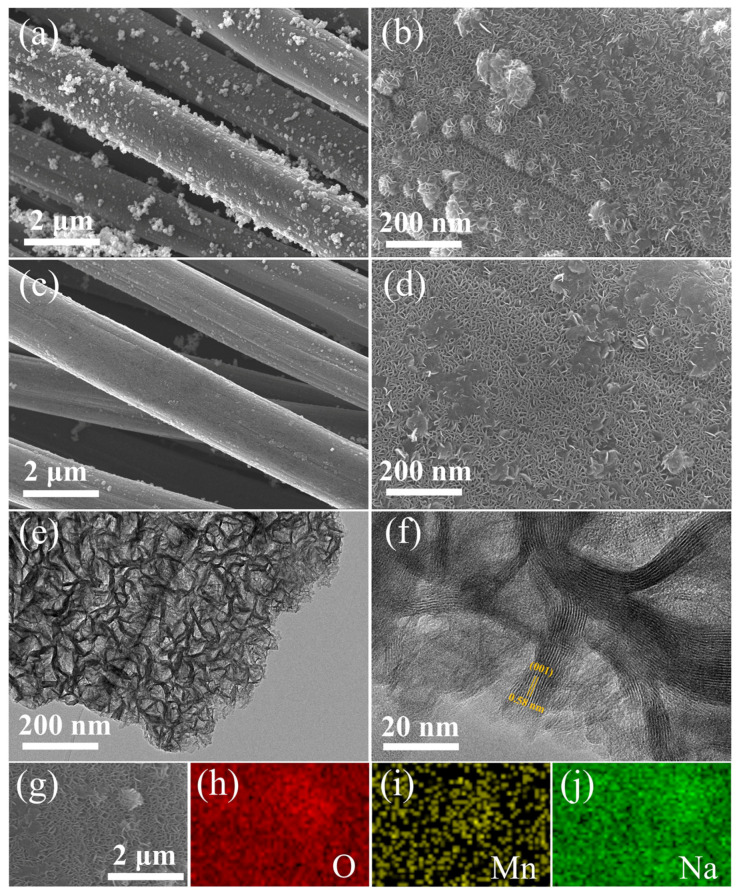
SEM images of (**a**,**b**) pure MnO_2_, (**c**,**d**) Na-MnO_2_. (**e**) TEM and (**f**) HRTEM images of Na-MnO_2_. (**g**–**j**) Corresponding area EDS mapping of Na-MnO_2_.

**Figure 2 nanomaterials-13-01075-f002:**
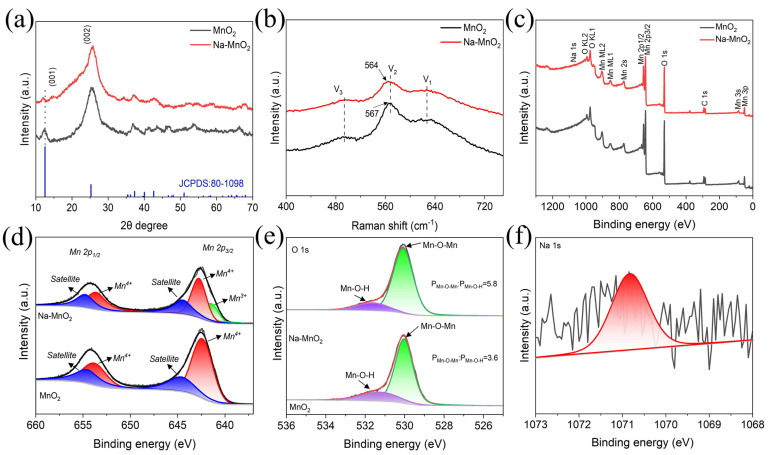
(**a**) XRD spectra, (**b**) Raman spectra, (**c**) XPS survey spectra of MnO_2_ and Na-MnO_2_ samples. (**d**) core-level Mn 2p, (**e**) core-level O 1s of MnO_2_ and Na-MnO_2_ samples. (**f**) core-level Na 1s spectra of Na-MnO_2_ sample.

**Figure 3 nanomaterials-13-01075-f003:**
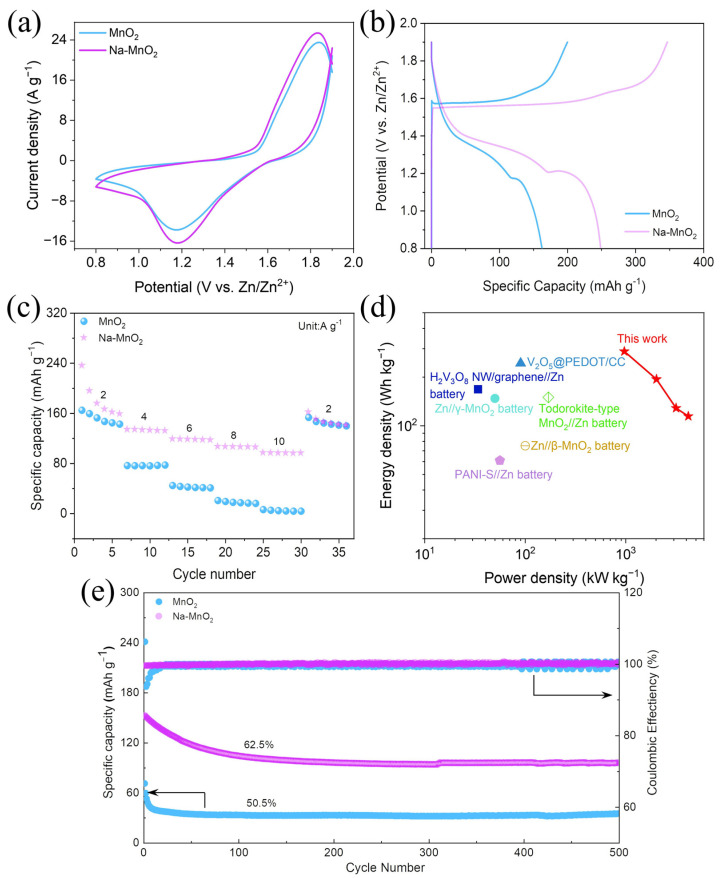
(**a**) The CV curves at a scan of 10 mV s ^−1^, (**b**) GCD curves at a current density of 2 A g^−1^ of MnO_2_ and Na-MnO_2_ samples. (**c**) Rate capacities of MnO_2_ and Na-MnO_2_. (**d**) Ragone plots of the Na-MnO_2_//Zn battery and other reported batteries. (**e**) Cycling stability of the MnO_2_//Zn and Na-MnO_2_//Zn batteries at 8 A g^−1^.

**Figure 4 nanomaterials-13-01075-f004:**
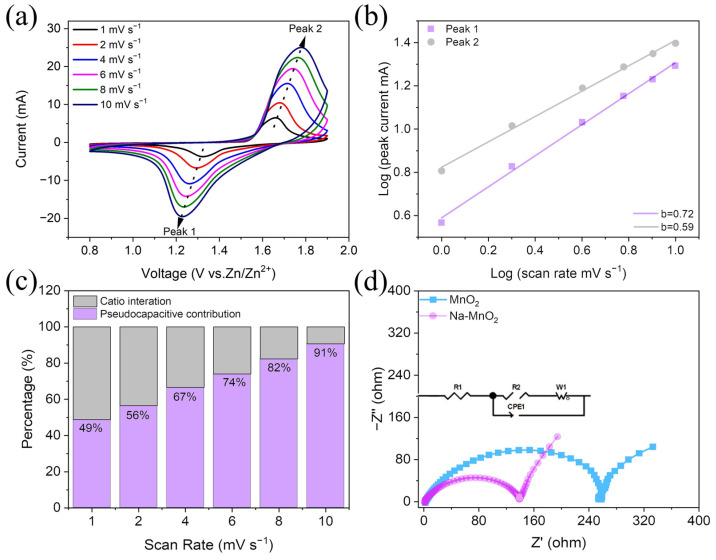
(**a**) CV curves of Na-MnO_2_ electrode at 1, 2, 4, 6, 8 and 10 mV s^−1^. (**b**) log (peak current) versus log (scan rate) plots of each peak. (**c**) The capacitive contributions of Na-MnO_2_. (**d**) Impedance spectra of MnO_2_//Zn and Na-MnO_2_//Zn batteries.

**Table 1 nanomaterials-13-01075-t001:** The energy density and power density of the Na-MnO_2_//Zn battery and other reported batteries.

ZIBs	Power Density (W kg^−1^)	Energy Density (Wh kg^−1^)	Ref.
V_2_O_5_@PEDOT/CC//Zn battery	90	243.3	[41]
Zn//β-MnO_2_ batter	75.2	100	[22]
H_2_V_3_O_8_ NW/graphene//Zn battery	168	34	[42]
Zn//δ-MnO_2_ battery	320	110	[20]
Todorokite-type MnO_2_//Zn battery	170	150	[43]
PANI-S//Zn battery	56	61	[44]
This work	967	289	

## Data Availability

The data presented in this study are available on request from the corresponding author.

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
