# Peer review of "Sodium Ion Pre-Intercalation of δ-MnO2 Nanosheets for High Energy Density Aqueous Zinc-Ion Batteries"

_nanomaterials, 2023, doi:10.3390/nano13061075_

Round 1

Reviewer 1 Report

The authors describe an ion pre-intercalation water bath method for the growth of (Na-MnO2), nanosheets on a flexible carbon cloth substrate to serve as a cathode for an aqueous Na-MnO2//Zn battery. Their study has shown that pre-intercalation of alkali cations is an effective method for improving the properties of MnO2/zinc storage devices and provides new insights for the development of flexible electrodes with high energy density.

The text contains a number of inaccuracies and ambiguities that need to be corrected and clarified. I list some of them:

Line 19 and throughout the text: Nowhere is it stated in what ratio the density is measured. In relation to the mass of the electrode?

Lines 140-142: There is no scale mark in Figure S1b, so it is not possible to derive the lattice constant. Also, this figure belongs to the manuscript.

Lines 154-155: The description of the figure is too short, and the same is true for the text, so it remains rather unclear what, for example, Figure 2f shows.

Lines 160-162: The resolution of Figure 2a is so low that the shift of the diffraction maximum to the left cannot be determined with certainty. Also, the dotted vertical line is not in the center of the MnO2 maximum.

Lines 175-176: Table S1 is not a proof. It is necessary to describe how the result of 18% sodium content was obtained.

Line 178: It is not clear what this valence state is and in what units it is measured.

Lines 180-181: In Figure S2 you can see something quite different, namely that the distance between the maxima of MnO2 (dashed vertical lines) is greater than the distance between the maxima for Na-MnO2.

Line 188: Can you really determine the binding energy (641.58 eV) to five significant digits?

Lines 193-194: It is not clear what the red curve in Figure 2f represents.

Line 203: Batteries cannot be measured, but some properties of batteries can be measured.

Line 209: The abbreviation GCD is not defined anywhere in the text.

Line 210: The1. 2V plateau probably corresponds only to the reduction peak.

Lines 231-235: This data would be much clearer in a table.

Figure 4b. Log should be written instead of lg in the axis label.

Lines: 274-277: It is not clear what the slope at low frequencies refers to. If it is a Warburg element in the Nyquist diagram (Fig. 4d), it corresponds to mass transport within the electrodes and should be discussed in more detail. Judging from the curvature of the curve, it is probably due to the finite length diffusion.

Author Response

Dear Reviewers,

Thanks very much for your efforts in the review of our manuscript entitled “Sodium Ion pre-intercalation of δ-MnO2 nanosheets for high energy density aqueous zinc-ion batteries” (nanomaterials-2252947). We deeply appreciate referees’ comments and constructive suggestions, which are valuable for improving the quality of our manuscript. We have carefully carried out additional experiments and improved the manuscript as referees suggested. Additionally, the point-to-point responses to the reviewers’ comments have been enclosed. We believe that the responses have addressed the suggestions and concerns of the reviewers. Please see the attachment.

Thank you for your consideration!

Sincerely,

Dr. Fuxin Wang

Reviewer 2 Report

Na-MnO2 electrode materials were prepared and characterized by XRD, SEM, TEM, Raman, XPS investigations. The electrochemical performances of this material as battery were demonstrated by cyclic voltammetry, electrochemical impedance spectroscopy and constant current charge/discharge. The paper can be published after the operating of following observations:

1.       The H2SO4 concentration used in the synthesis will be indicate.

2.       XRD data - The larger diffraction peak centered at about ~25o (carbon ???) is not explicated.

3.       Raman Spectra – The decrease of intensity of the Raman band noted V2 will be detailed by the intercalation of Na+ ions.  

4.       XPS data – A detailed interpretation of the XPS data regarding the modifications of position and intensity of Mn2p and O1s spectra by doping will be presented.

5.       The word “thesis” will be substituted by “paper or work”.

Author Response

(The authors gave the same response as above.)

Reviewer 3 Report

In this paper, the authors present an interlayer modulation strategy using the water bath method to create Na-MnO2 nanosheets that can serve as electrode materials. By pre-intercalating Na+, they were able to increase the layer-to-layer distance of δ-MnO2 nanosheets, improving their ion transport rate and stability. The authors fabricated a Na-MnO2//Zn battery with a Na-MnO2 cathode and a Zn plate anode, achieving a high specific capacity of 251 mAh/g at 2 A/g, with excellent rate performance (96.9 mAh/g at 10 A/g). Additionally, the battery maintained 62.5 % of its initial capacity after 500 cycles and had almost 100% Coulomb efficiency. Furthermore, the battery exhibited impressive energy density (289 Wh/kg) and peak power density (0.967 kW/kg). In my opinion, this work has great potential for the development of manganese-based high-energy density flexible electrodes. Therefore, I highly recommend accepting this article in its current version.

Author Response

Dear Reviewers,

Thanks very much for your efforts in the review of our manuscript entitled “Sodium Ion pre-intercalation of δ-MnO2 nanosheets for high energy density aqueous zinc-ion batteries” (nanomaterials-2252947). We deeply appreciate referees’ comments and constructive suggestions, which are valuable for improving the quality of our manuscript. Thank you very much for your positive comments. We greatly appreciate your efforts on our manuscript.

Thank you for your consideration!

Sincerely,

Dr. Fuxin Wang

Round 2

Reviewer 1 Report

The authors have generally accepted all the comments and made the necessary corrections.

Only a few comments remain:
- It would be useful if they would add somewhere at the beginning of the text a sentence like The energy density and the power density is calculated with respect to the mass of the active material.
- Instead of average valence state (AOS), it should say average oxidation state (AOS).
- After adding the table, lines 244-247 become redundant.

Author Response

Thank you again for your positive comments. We have revised our manuscript according to your suggestions. Details, please see the attachment.
